# Effect of Regulated Deficit Irrigation on Agronomic Parameters of Three Plum Cultivars (*Prunus salicina* L.) under Semi-Arid Climate Conditions

**DOI:** 10.3390/plants11121545

**Published:** 2022-06-10

**Authors:** Hichem Hajlaoui, Samira Maatallah, Monia Guizani, Nour El Houda Boughattas, Anis Guesmi, Mustapha Ennajeh, Samia Dabbou, Félicie Lopez-Lauri

**Affiliations:** 1Regional Center for Agricultural Research of Sidi Bouzid, Sidi Bouzid 9100, Tunisia; hajlaoui2001@yahoo.fr (H.H.); samiraham2003@yahoo.fr (S.M.); anis.gasmi@iresa.agrinet.tn (A.G.); 2Laboratory of Non-Conventional Water Valuation (INRGREF), University of Carthage, Carthage 1054, Tunisia; 3Laboratory of Remote Sensing and Information Systems with Spatial Reference (LTSIRS), National Engineering School of Tunis (ENIT), University of Tunis El Manar, Tunis 1068, Tunisia; nourelhoudaboughattas@yahoo.fr; 4Unit of Biodiversity Research and Valorisation of Bioressources in Arid Regions, Faculty of Science of Gabes, University of Gabes, Cité Irriadh-Zrig, Gabès 6072, Tunisia; ennajehmustapha@yahoo.fr; 5Unit of Bioactive and Natural Substances and Biotechnology UR17ES49, Dentistry Faculty, University of Monastir, Monastir 5000, Tunisia; dabbou_samia@yahoo.fr; 6UMR 95 Qualisud, Laboratory of Physiology of Fruits and Vegetables—Campus Agroparc, University of Avignon Pole Agrosciences, 301 Baruch Spinoza Street, 84916 Avignon, France; felicie.lauri@univ-avignon.fr

**Keywords:** moderate stress, plum fruits, tree growth, photosynthesis, water use efficiency

## Abstract

Regulated deficit irrigation (RDI) strategies may greatly contribute to save irrigation water, especially in low water availability conditions. The effects of RDI on the growth process, photosynthesis, fruit yield, and some quality traits were assessed for two years on three plum (*Prunus salicina* Lindl.) cultivars (‘Black Diamond’, ‘Black Gold’ and ‘Black Star’) grown in Midwest Tunisia. The experiment was conducted during two successive seasons (2011–2012). Two water regimes were applied per cultivar during the phase of fruit growth until fruit ripening. Stressed trees receive 50% of the amount of irrigation compared to controls (CI). Several eco-physiological parameters and some pomological criteria were measured, based on the climatic condition (ETo, ETc, and VDP). Results showed that the three cultivars had an aptitude for tolerance for moderate stress with varying degrees of response time to drought stress. Globally, a slight decrease in the gas exchange rate (AN and gs) and the water status (RWC, Ψmin, and Ψos) was registered. Tree yields, size and weight show a slight decrease under water stress. Nevertheless, there was an improvement in the total soluble solid content (SSC) and firmness in same cultivars. Our results proved that the Black Star cultivar was the most tolerant to deficit irrigation, in reason that it maintains a good water status and a high photosynthetic activity.

## 1. Introduction

Water scarcity is considered the world’s largest environmental stress on agricultural production worldwide [1]. Global average temperatures were anticipated to rise in the coming years, resulting in greater evaporation rates [2], and leading to an increase in crop demand for water irrigation. The water consumption of agriculture in the Mediterranean region dissipates around 177.7 109 m^3^ year^−1^ and good management of irrigation would reduce this loss of water resource [3]. Water scarcity dominates restrictions on the production of important crops essentially in the Mediterranean-type eco-systems [4]. In Tunisia water limitation particularly affects arid and semi-arid regions, which are located the important arboriculture areas. Therefore, water resources available for agriculture have to be used more efficiently. In this regard, regulated deficit irrigation (RDI) is widely used in fruit tree orchards. This practice is based on the application of amounts of water less than the full crop water requirements at the selected time. It is a key tool for optimizing water use efficiency. RDI was suggested around early 1980 in peach trees [5] and it has been usual research interest in most fruit trees [6]. For instance, regulated irrigation deficiency has been projected to limit fruit orchard water consumption and encourage plant adjustment to dry stress [7].

In recent years, Tunisia has been affected by several episodes of reduced rainfall [8]. This water shortage has led to a significant reduction in fruit trees production [8]. It has even caused the death and loss of several cultivars, especially in the mid-west and southern regions of the country. The restoration of affected orchards and the creation of new ones have made it possible to identify the most strategic resistant local cultivars, investigate exotic cultivars, and define appropriate new practices [9].

Drought has created profound consequences in the physiological function of the plum tree, worldwide, and it affects particularly its growth and productivity. The physiological processes of plants, such as gas exchange parameters (photosynthesis and transpiration), depend on the promptness, severity, and extent of the drought episode [10]. In general, the first sign of water deficit in the plant becomes obvious at the stomatal level. Indeed, stomata reduce their degree of opening to prevent dehydration [11]. Subsequently, photosynthesis is influenced by internal water limitation that is following stomatal closure and the consequent decrease in CO_2_ availability at the level of the chloroplast.

Promoting productivity during water deficit conditions is the main objective of plant breeding [1]. In such context, to adapt to arid soil, plants performed much behavior among which we can find physiological, reactions as well as ecological strategies, by either stress prevention or stress tolerance [12]. Nevertheless, the response of plants is mainly related to the type of water shortage that brings physiological changes [13], the adaptation to a certain level of water accessibility, and the readjustment to drought [14]. Understanding the water deficit resistance processes facilitates the application of deficit irrigation strategies designed to save water and reduce the side effects on crop productivity [15]. The resistance of plum trees to water stress is linked to internal adaptive capacity processes that occur in their leaves or roots. A lowering of osmotic potential, stomatal conductance and transpiration decrease, leaf shedding, and the ratio of the dry weight root to shoot can manifest these mechanisms [16]. The quality of plum fruits is an essential element for its market value. An imprecise irrigation process can lead to overexploitation of water resources and low fruit quality as the decrease in size and weight. The impacts of deficit irrigation (DI) on reducing water consumption, yield, and quality in various crops and fruit trees have been widely studied since the 1980s [17]. Also, since the1990s, the effect of deficit irrigation on the fruit quality and the associated index of soil water deficit are investigated from qualitative depiction to quantitative index [18]. Some researchers have confirmed that DI improves the quality of fruits, namely the physical and chemical characteristics of the fruit [19]. A previous study had shown that deficit irrigation increases the level of sugar in peach fruits [20]. Furthermore, another study proved that water deficit increased the level of phenolic compounds and improved the antioxidant activities [21].

The effect of water deficit leads to a decrease in photosynthesis resulting in a reduction of the carbohydrate intake towards the fruits. On the other side, it also creates adequate conditions favorable to oxidative stress, which is accompanied by the production and accumulation of reactive oxygen species (ROS), including O_2−_, •OH and possibly H_2_O_2_ [22]. Since oxidative stress incites the accumulation of antioxidant components, so water deficit could generate a positive effect on fruit value to human health. Generally, some recent studies [23,24] have highlighted some specific patterns of crop quality due to several abiotic stressors and claimed a predisposition for loss of taste but an improvement in nutritious value. [25] Previous work has shown that deficit irrigation applied during the period of fruit development and/or after harvest of peach tree considerably reduced vegetative growth; however, there was no effect on the fruit production until the fourth year. Furthermore, deficit irrigation can also speed up the ripening of the late-maturing peach (*Prunus persica* L.) by one week [25]. Indeed, the early ripening of fruits is advantageous in increasing their market value. Understanding the physiological responses of different plum cultivars to moderate water stress is a key process not only for determining threshold stress levels but also for selecting sensitive and resistant genotypes for plum breeding programs. The present study aimed to evaluate the effect of regulated water stress in semi-arid climate conditions in three plum cultivars and their effect on photosynthesis, and water status to determine the most resistant cultivars and to investigate the role of the regulated deficit irrigation (RDI) regime in the improvement of plum fruit quality. For this purpose, the determination of physiological and biochemical parameters of three plum cultivars (Black Diamond, Black Gold, and Black Star) subjected to two different irrigation strategies (CI and RDI) was conducted during two consecutive seasons.

## 2. Results

### 2.1. Environmental Conditions

Environmental conditions during the experimental period are shown in Figure 1 and Figure 2. In 2011 and 2012, the site was characterized by scarce precipitation in the period before starting deficit irrigation, January–April (5 and 19 mm, respectively) while rainfall during May–August was almost zero. Irrigation volume was comparable between the two seasons. Further, the temperature was very high and varied between (39 and 46.6 °C) in May and July successively. The soil composition of the orchard is indicated in Table 1.

### 2.2. Tree Growth Pattern

The effect of regulated deficit irrigation (RDI) on the tree growth pattern of the three plum cultivars was assessed by measuring the shoot growth. There were significant differences among the three cultivars. Our data (Figure 2) showed also that, during the whole study period, RDI affected considerably the shoot growth (*p* < 0.05). Globally, BD and BS cultivars have shown the same behavior of shoot growth. Indeed, the two cultivars displayed a significant decrease in their growth during June, July, and August. However, this decrease was found to be more pronounced in BS cultivar than in the BD cultivar. For the latter cultivar, the reduction rate in shoots growth during July was approximately 64% and 61% in control and stressed plants, respectively. On the other hand, BG showed completely different behavior. In the control plants, shoot growth showed a slight increase mostly during June and July (the increase was about 12% during July) while there was an important reduction only in August (a drop of 54%). For stressed plants, the growth of branches decreased in June and August (Figure 2).

### 2.3. Water Status

The water status of plum cultivars studied was characterized by leaf water potential (Ѱmin) (Table 2). Before the application of RDI treatment, Ѱmin was nearly equal in both control and stressed tree. During the study period, midday leaf water potential (Ѱmin) in controls treatment (phase III of fruit growth) was maintained between −1.6 MPa and −3.1 MPa for 2011 and 2012. However, in all cultivars subjected to the water deficit, Ѱmin become more negative and reached −3.6 MPa. For cultivars BD and BS, the reduction of Ѱmin was significant, particularly in the last week of July; it was varied respectively, from −1.7 MPa to −2.6 and −2.2 MPa to −3.1 MPa. In the same period, BG has a slight reduction of Ѱmin. It varied from −2.1 MPa to −2.6 MPa. The osmotic potential is a major component of the water potential. In this context, our results showed that in control trees, leaf osmotic potential (Ψos) decreased slightly over time. It was nearly stable between −1.40 MPa and −1.65 MPa in all the cultivars (Table 2). The application of moderate water stress caused a drop in Ѱos in stressed trees for all cultivars studied. In fact, Ѱos dropped to −2.22 and −2.38 MPa in BD and BS respectively and to −2.58 MPa in BG, while the rate of this reduction depends on the cultivar. The decrease was more pronounced in the BS cultivar with approximately, 42%. While it was lower in BD and BG with nearby 32% and 36%, respectively.

Relative water content (RWC) is one of the preliminary parameters that characterized water status in the trees. In plums, moderate water stress caused a slight decrease in leaf relative water content of the tree (Figure 3). Nevertheless, the intensity of this reduction depended on the cultivar. In control trees of all cultivars, RWC had decreased slightly over time. While in stressed trees, the diminution was more significant. In BD cultivar RWC, declined by approximately 13% between the beginning and the end of the study period (15/6–27/7) as shown in Figure 3. This drop is about 10% during the same period for the cultivar BG. The cultivar BS was less influenced by water treatment and its RWC decreased only by 6% (Figure 3).

### 2.4. Gas Exchange

During the application of RDI treatment, the mean values of the net CO_2_ assimilation rate (AN; μmol CO_2_ m^−2^ s^−1^), and stomatal conductance (gs, mmol H_2_O m^−2^ s^−1^), showed a significant decrease compared to control plants. Indeed, in all cultivars and stressed plants, the net CO_2_ assimilation (AN), varied between 11 and 15 μmol CO_2_ m^−2^ s^−1^. However, in control one, the values reached 35 μmol CO_2_ m^−2^ s^−1^ (Figure 4). At the end of the study period, deficit irrigation reduced AN in all stressed trees differently between cultivars. In fact, in both cultivars, BD and BS the reduction of AN was about 53% compared to controls while in BG, the reduction of AN was only around 37% compared to controls. The same thing for gs in fact by the end of July stressed trees from the cultivars BD and BG increases their stomatal conductance to 288.9 and 262.1 mmol H_2_O m^−2^ s^−1^, respectively. However, in control trees, gs was 209 and 203 mmol H_2_O m^−2^ s^−1^ for BS and BG, respectively for the same period (Figure 4). In the three studied varieties BD, BG, and BS, and during the experiment period, net assimilation of CO_2_ was higher in controls trees compared to the stressed. Controls trees of BS cultivar had greater photosynthetic activity in May (26.87 in BS compared to 18.7 and 18.03 μmol CO_2_ m^−2^ s^−1^ in BD and BG, respectively) and remained constant throughout the study period with a net increase in August (33.16 μmol CO_2_ m^−2^ s^−1^). The net assimilation of CO_2_ did not vary significantly in BD and BG. While AN decreases significantly during June at BS. 

The intrinsic water use efficiency (WUEi = AN/gs) in three plum cultivars studied varied significantly among cultivar and irrigation treatment applied. Indeed, throughout the experiment, the WUEi for the two cultivars BD and BG remained more important in control trees than stressed trees. For example, for BG, WUEi increased from 40% in May to 61% in August (Figure 4). The BS variance showed a different behavior, the WUEi values recorded an increase in the stressed trees compared to the control trees just in July. This rise was about 22.4%.

### 2.5. Correlations

Stomatal conductance showed a strong correlation with predawn leaf water potentials (Figure 5) with a highly significant slope different from zero and an adjusted r^2^ = 0.92 for control irrigation and r^2^ = 0.81 for regulated deficit irrigation in the leaves of BD cultivars.

Our results showed also, a linear and positive correlation (r^2^ = 0.96) in the control treatment. The opposite is observed in the control trees of the Black Star variety (r^2^ = 0.41), where WUE was negatively linearly correlated with VPD. But this is not the case for the other varieties, which did not show any significant correlation between these two parameters (Figure 6).

### 2.6. Plum Yield and Fruit Quality Evaluation

The fruit yield (FY) expressed as the fruit production per unit of one tree showed that the BS cultivar had the most important FY (100 kg tree^−1^) while, the BG cultivar had the lowest fruit yield (86 kg tree^−1^) (Table 3). Our results showed that irrigation treatments had affected fruit yield. FY was significantly lower in the regular deficit irrigation treatment than in the full irrigation treatment, also BG cultivar was the most affected by deficit irrigation (5 kg tree^−1^), while BS was the least affected (3 kg tree^−1^). Moreover, Table 3 illustrated the effect of water deficit on the plum morphometric characteristics (size and weight). Our results indicated that water deficit affected the average fruit diameter. Furthermore, intraspecific differences were observed. Indeed, in BS fruits were the less affected by the water shortage compared to controls where the reduction was only about 2–3 mm. However, in BD cultivars, fruits were the most affected.

Our results showed that moderate water deficit decreased significantly fresh weight in the different plums cultivars studied (Table 3). Indeed, in all treatments, the BS cultivar showed the heaviest fruits weight followed by BD than BG. Moreover, compared to controls, the reduction of fruits weight induced by water deficit was less important in the BS cultivar (about 13%) than in the two other cultivars BD and BG.

Furthermore, the results indicated that water deficit had a negative effect on the titratable acidity of plums juice (Table 3) which had decreased significantly in the three cultivars studied. Under controls conditions, the BD cultivar showed the highest acidity (3.96% malic acid), while BS fruits had the lowest one (3.65% malic acid). Our results showed, also, that, at harvest, soluble solids content had increased under water deficit in the different cultivars studied (Table 3), and the increase had varied depending on cultivars. Indeed, the BD cultivar had the highest °Brix of juice, for which the soluble solids content passed from 16.66% for the controls to 17.80% in the stressed. However, the BG cultivar, which had the lowest total soluble solid content, showed a slight increase from 14.66% to 14.93%. In addition, deficit irrigation treatment had influenced significantly (*p*< 0.001) and positively the fruit firmness. In fact, in the three cultivars, fruits from stressed trees were firmer than controls fruits (Table 3). Among varieties, fruits from the BG cultivar revealed a significantly higher firmness followed by BD fruits.

## 3. Discussion

This study examined the stress effect on the shoot growth of the three plum cultivars. The BD and BS cultivars have the same shoot growth behavior. Indeed, these two cultivars displayed a significant decrease in their growth during June, July, and August. Moderate water deficit applied to three plum cultivars studied influenced their water status. The midday water potential decreased since trees demonstrated various dry stress responses compared to control trees. The reduction in midday water potential was less important in BG cultivar at the end of July (the hottest month of the experimental period). Similar results were found in plum [26], peach [27], and mango [28]. On the other hand, this reduction was more intense in BD and BS during July (Table 3. The application of moderate water stress caused a drop in Ѱos in stressed trees of three cultivars. Ѱos dropped to −2.22 and −2.38 MPa in BD and BS respectively, and to −2.58 MPa in BG. However, the amount of this decrease depends on the cultivar. The decrease was more pronounced in the BS cultivar with approximately 42%. While it is lower in BD and BG with around 32% and 36%, respectively. The relative water content is an appropriate parameter to characterize the physiological water status of a tree. The results obtained in our study showed that among the three cultivars moderate water stress reduced leaf RWC. However, the RWC was the least affected compared to BD and BG. This inter-varietal variability was also distinguished in olive [29] and almond trees [30].

The tolerance to water stress is related to the capacity of mixing efficiently osmotic adjustment with stomatal control processes to permit sustained growth. Water deficit has a direct effect on the photosynthetic activity of the plant. It required the gradual closing of the stomata, which in turn reduced gas exchange (photosynthesis and transpiration) [31]. The results of our study showed that moderate water stress caused a reduction in photosynthetic assimilation and stomatal conductance in the three plum cultivars. However, the cultivar BD appeared to be the least affected. It exhibited a recovery at the end of the treatment period. Our results are consistent with those found in peach [32,33], vine [34], and mango [35]. Stomatal conductance (gs) is considered the main factor controlling photosynthesis assimilation in mango [36] with several studies indicating highly significant linear or exponential relationships between gs and AN. Those last decreased significantly in stressed trees of three cultivars compared to the control (CI). This response reveals a strong stomatal adjustment to prevent extreme water loss under RDI (50%). This comportment has been marked also in water-stressed in almond trees [37] and also in other fruit trees, such as *Prunus persica* L. [20]. The changes in stomatal conductance (gs) in response to environmental abiotic stress represent the primary physiological signal that plants regulate gas exchange and water flow via the soil-plant-atmosphere continuum over the short term.

There was also a robust relationship between stomatal conductance and predawn potential (Ψmin) in the other cultivars studied. The relationships between stomatal conductance (gs), leaf water potential (Ψmin), and environmental variables are complex. The feedback mechanisms between these variables and differences between cultivars have been reported previously [38,39]. Our results showed that at the beginning of the water stress application period, there was a significant decrease in WUEi in the BD and BS cultivars, especially in the last one. When the water deficit increases and persists, metabolic limitations of photosynthesis settle. This is reflected by the decrease in WUEi [40]. WUEi decreased in stressed trees of three cultivars studied to become lower than that of controls. Except for the BS cultivar, of which WUEi increased considerably during the month of July, this increase was in order of 22.4%. This was reliable to the progressive increases in AN/gs as drought progresses. The decrease in AN, E (data not shown in this article), and gs of CI and those of RDI was caused by climate changes such as air temperature and pressure deficit in water vapor. As vapor pressure deficit between leaf and air increases, stomata usually react through partial close. In frequent cases, stomatal conductance decreased greatly with growing VPD. This may explain the high correlation reported WUE with VPD (Figure 6). This correlation is consistent with our measurements exclusively in the variety Black gold (BG).

Our results displayed that, in the three cultivars studied, fruit yield was affected by RDI (Table 2). The negative impact of RDI in yield response could be caused by (i) the significant decrease of photosynthetic capacity of the leaves exposed to water deficit during water stress, which may result in a reduction of assimilates offered for fruit growth [41], and (ii) a stronger contest for assimilates between sink organs. Some authors attribute the main loss of yield in dry conditions to the reduction of photosynthesis alone [42]. The present study showed that in three plum cultivars studied, the use of moderate water stress during fruit development and maturity phase reduced juice, which is in agreement with other works on fruit species [43]. The drop in fruit acidity during maturation is in agreement with a previous study [44], it can be due to the decrease of malic acid, the main organic acid in plums, which makes most of a titrable acidity (Table 3). The reduction of the malic acid concentration is generally attributable to its use in Krebs in respiratory cycles or gluconeogenesis and anthocyanin synthesis. During the fruit development period, an overall irrigation reduction had significantly increased SSC in BD cultivar; however, no significant difference was recorded for BS and BG (Table 3). Sugars are the main constituent of SSC. Previous works suggest that, RDI improves fruit quality during fruit development by increasing sugar content [18,24,45]. Whereas, in this study the slight increase in SSC under deficit irrigation might be explained by the fact that the fruit had likely less water, causing the carbohydrate content to seem greater. Indeed, the decrease of fruit size and juice content in fruit exposed to water deficit probably confirm this suggestion. In this context, some authors [46] reported that the water content influences carbohydrate concentration in the early stages of fruit growth through both metabolism and dilution/dehydration effects, but mostly through dilution/dehydration effects in the late stages of fruit development.

The growth of fruit under water stress of the three cultivars studied has also been previously reported [20] and accredited to increased cellular density in water-stressed fruit because of a decrease in fruit size [47]. For example, RDI raises the firmness of pears [48] but, no effect was observed for apricots *Prunus armeniaca* L. [49]. Therefore, there have been certainly other elements influencing firmness in addition to fruit density and size. The decrease of cellular hydration as an outcome of water stress can also bring flesh firmness increase [47]. Regulated Deficit Irrigation has an impact on fruit production and chemical composition. In this work, water stress in the three plum cultivars reduces the rate of juice. Black Diamond fruits were the most affected with an average reduction of 6.41% compared to the control treatment. Many researchers proved the existence of relations between fruit size and water stress. In addition, [48] demonstrated a similar effect with a dropped fruit weight and a minor increase of fruits number per tree, despite the absence of significant differences, which agrees with previous work in peach fruits [20]. These results could be explained by the fact that water increased the soluble solids content [49] and at the same time reduced the size of fruits [50] resulting in a decrease in the juice content. Indeed, water deficit did not affect the dry weight of fruit, but it reduced its weight [51]. In another study, [52] claimed that water stress results in the increase of TSS and TA, yet this was not the outcome of fruit dehydration but rather of the osmoregulation that was carried by water shortage.

## 4. Materials and Methods

### 4.1. Plant Material, Experimental Conditions and Irrigation Treatments

The present work was conducted during two successive seasons in a commercial plum orchard located 5.28 km north of Regueb city (Midwest of Tunisia) (34°50′42,23″ N, 9°50′21,61″ E) and 160 m above sea level. The region is characterized by silty clay loam soil and situated in a semi-arid bioclimatic featured by low rainfall and hot long summer (Figure 1 and Figure 2). The study was focused on three plum cultivars (‘Black Star’ (BS), ‘Black Diamond’ (BD), and ‘Black Gold’ (BG)), grafted on the ‘Mariana’ rootstock, planted in 2006 year at a spacing of 4 m × 5 m. All cultivars received drip irrigation with two drippers per row and 4 L h^−1^ micro-sprinklers for each irrigated tree. During the two years of experimentation, all cultivars received the same amounts of nitrogen and potassium fertilizers. The study period was running from March to July. Some meteorological parameters describing the conditions of the agricultural season are presented below (Figure 1A,B). The experimental period was characterized by low rainfall and high temperatures, which increased the trees’s evapotranspiration.

For the application of water deficit, treatments were distributed randomly in the three experimental blocks. In each block, we found four lines of the same cultivar with 18 trees in each line. Two lines were used as a control treatment (CI) while the two other lines were exposed to regulated deficit irrigation (RDI). The controls received full replacement crop evapo-transpiration (ET_c_) throughout all the developmental phases:

ET_c_ = ET_0_ × Kc, (Figure 1) data obtained from MODIS (ftp://e4ftl01.cr.usgs.gov/MOLT/MOD11C3.005) accessed on 26 March 2022 [53]. where ET_0_ is the reference evapotranspiration and Kc is the crop coefficient. During the experimental study, control trees received between 12–30 L day^−1^ depending on the season, meteorological data, and phenological stage. The trees exposed to RDI (regulated deficit irrigation) received 50% of the amount of control irrigation. RDI treatment was applied for three months (May-June-July) corresponding to the fruit growth period, which coincides with the dry season in the Regueb region. Three replications per treatment are considered. The water deficit implicated in the present study can be considered moderate. During the experimental period, several physiological and pomological parameters were measured at regular intervals (15 days).

### 4.2. Soil Analysis

Soil samples were taken with a soil probe to a profundity of 50 cm in the orchard. Total nitrogen was measured in a ‘Gerhardt’ digestion system. The potassium assay was performed by flame photometry and the detector used was a PFP manufactured by Jenway Ltd. (London, UK). Calcium and magnesium were determined by atomic absorption spectrophotometry in a Perkin Elmer 2380 spectrophotometer (Waltham, MA, USA) (Table 1).

### 4.3. Measurements of Shoot Growth

During the experimental period, shoot growth was assessed by measuring the shoot extension at different time intervals; to assess the impact of the different irrigation treatments on the behavior of the three plum cultivars studied (BD, BS, and BG). In the early 2011 and 2012 growing seasons, twelve shoots of the last season (1-year-old) per treatment. Measurements were carried out every 30 days. Shoot growth was determined on three trees per treatment by measuring shoots on four tagged shoots (*n* = 12).

### 4.4. Tree Water Status

Leaf water potential (Ψmin) was periodically measured in Four leaves taken from each tree and three trees were chosen for each treatment (*n* = 12) using a Scholander pressure chamber (PMS Instruments, Corvallis, OR, USA). The technique involves measuring the pressure of the xylem sap in the vessels of a shoot freshly picked [54].

The relative water content (RWC) was determined by the method of Kramer [12]. Briefly, four leaves from three trees were collected from each treatment, then immersed in distilled water for 24 h at 4 °C until saturation. Finally, the leaves were dried in an oven at 80 °C for 48 h.

The RWC was calculated using the following formula:RWC = (FW − DW) × 100/(FWsat − DW)

With:

FW: fresh weight;

DW: dry weight;

FW sat: fresh weight at saturation.

### 4.5. Osmotic Potential (Ψos)

Osmotic potential (Ψos) was measured according to the method described in previous work [55]. Four leaves from each tree (and three different trees were chosen for each treatment), (*n* = 12) were cut into a small piece and placed in an Eppendorf tube with four holes at its base. Next, the tube was immersed in liquid nitrogen for a few seconds, followed by 5 min of thawing (3 cycles of freezing and thawing are carried out). Then, the whole was centrifuged at 8000× *g* and a temperature of 4 °C for 15 min. The tissue sap collected was analyzed for Ψos estimation. Osmolarity (c) was assessed with a vapour pressure osmometer (Wescor 5500, Logan, UT, USA) and converted from mOsm kg^−1^ to MPa using the formula:Ψs (MPa) = −c (mOsm kg^−1^) × 2.58 10^−3^ according to the Van’t Hoff equation

### 4.6. Gas-Exchange MEASUREMENTS

The parameters of gas exchange were determined on four leaves from each tree and three trees for each treatment (*n* = 12), using an open portable system ADC model LCA-4 infrared gas analyser (Analytical Development Co., Hoddesdon, UK) operated at 200 mmol m^−2^ s^−1^ flowrate in conjunction with a portable temperature- and humidity controlled leaf chamber with a surface area of 6.25 cm^2^, under ambient temperature, relative humidity and full sunlight conditions (at 11:00 a.m.–01:00 p.m.). The CO_2_ concentration inside the leaf chamber was fixed at 338 mmol mol^−1^. Net assimilation (AN; µmol CO_2_ m^−2^ s^−1^) and Intrinsic water use efficiency (WUEi, mmol CO_2_ fixed mmol^−1^ H_2_O transpired; AN/gs) were calculated according to previous work [56].

### 4.7. The Yield Determination

At the end of each season, the yield was determined for each tree from the three cultivars (BG, BD, and BS). For each treatment, three trees were chosen by weighing the plum fruits, dividing the last final yield (kg tree^−1^) by the volume of water applied.

### 4.8. Pomological Parameters

#### 4.8.1. Sample Processing

The fruits were collected in July at the maturation stage. Just after harvest, 10 fruits per tree and three trees per treatment were studied for each cultivar (*n* = 30), as shown below. Fruits were tested for diameter, firmness, weight, soluble solid content (SSC), and titratable acidity (TA). For biochemical analysis, the fruit samples were frozen and milled in liquid nitrogen then, stored at −80 °C until analysis.

#### 4.8.2. Fruit Quality Parameters

Generally, fruit size and organoleptic property are the most essential parameters that contribute considerably to the market value of fruits. The width (cm) and fresh weight of each fruit were measured respectively using a caliper (Mitutoyo, Manchester, UK) and a precision balance. Flesh firmness was determined on a partially peeled fruit by a penetrometer (FT 327, Italy). The juice content was determined on three recently harvested plums per replicate. Fruit samples were pureed in a blender and their debris was removed by centrifugation at 1500× *g* at 4 °C for 15 min. Juice content was calculated by the following formula:Juice content (%) = [Weight of juice (g)/Total weight of fruit (g)] × 100(1)

The soluble solid content (SSC) of juice, was measured by a digital refractometer (Atago-Palette PR 101; Atago Co., Tokyo, Japan) and SSC was expressed as °Brix. The pH of the juice was measured by a pH meter (MP 220, Mettler Toledo, Greifensee, Switzerland). Multiple readings per sample are made. Titratable acidity was measured in plum juice diluted in water by neutralizing free acids with a solution of 0.1 N NaOH added dropwise until pH 8.2. The results were expressed as % malic acid, the major acid in the plum.

### 4.9. Statistical Analysis

The experiment was conducted during two consecutive seasons (2011 & 2012). Results are presented as the average of three replications. Statistical analyses (ANOVA, Duncan test) were performed by the SPSS statistical software (version 13 for Windows, Chicago, IL, USA). 

## 5. Conclusions

In summary, the results for the whole studied period showed that the decrease in shoot growth was more pronounced in Black Star cultivars in comparison with Black Diamand and Black Gold. The application of moderate water stress caused an intense drop in the midday water potential, Ѱos, and RWC in stressed trees of three cultivars mostly in BS. RDI caused a reduction in gas exchange parameters (photosynthetic assimilation and stomatal conductance) in the three plum cultivars. Nevertheless, the BD cultivar was the least affected. Stomatal conductance showed a strong correlation with predawn leaf water potentials with a highly significant slope different from zero in the three cultivars and under both water regimes. WUEi decreased in stressed trees of three cultivars studied to become lower than that of controls (except the BS cultivar, where WUEi increased considerably during July).

In addition, moderate water deficit application during the fruit maturity period reduced the titratable acidity of plum juice and increased SSC in some cultivars. We also, found that fruit yield was affected by RDI in the three cultivars studied. According to our results, it is clear that the Black Star cultivar displayed a special behavior compared to the other cultivars. The physiological behavior concerning the RDI leads to a qualitative improvement of the plums. Physiological and organoleptic behavior approved that the Black Star cultivar is more appropriate to the central-western region of Tunisia (semi-arid climate).

## Figures and Tables

**Figure 1 plants-11-01545-f001:**
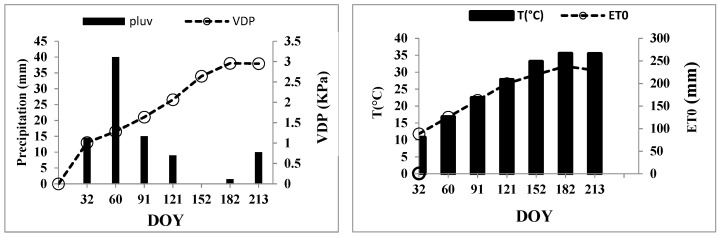
Precipitation (mm) and vapor pressure deficit (KPa) in terms of DOY. Average of daily mean air temperature (T) and crop reference evapotranspiration (ET_0_) in terms of DOY, for the period 2011–2012. The meteorological station is located 1 km from the experimental orchard.

**Figure 2 plants-11-01545-f002:**
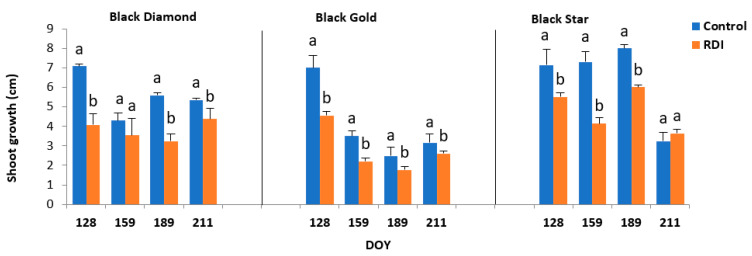
Shoot growth (cm) of three plums cultivars Black Diamond (BD), Black Gold (BG), and Black Star (BS) subject (Regulated Deficit Irrigation) or not (Control Irrigation) to deficit irrigation. Values are the means of twelve samples (*n* = 12) ± standard deviation. The letters a and b indicate significant differences (*p* ≤ 0.05) between the two irrigation treatments, within the same date.

**Figure 3 plants-11-01545-f003:**
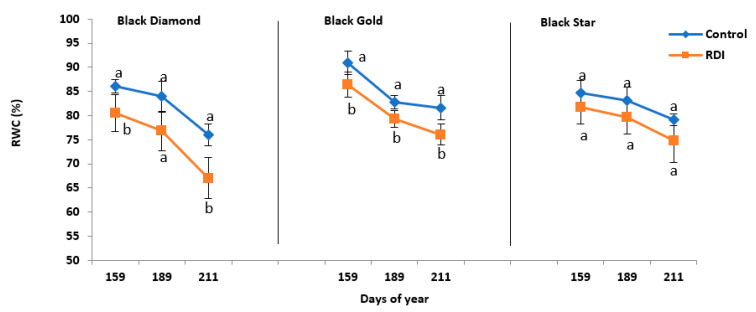
Relative water content (RWC %) of three plums cultivars BD, BG, and BS subject (RDI) or not (CI) to regulated deficit irrigation. Values are the means of twelve samples (*n* = 12) ± standard deviation. The letters a and b indicate significant differences (*p* ≤ 0.05) between the two irrigation treatments, within the same date.

**Figure 4 plants-11-01545-f004:**
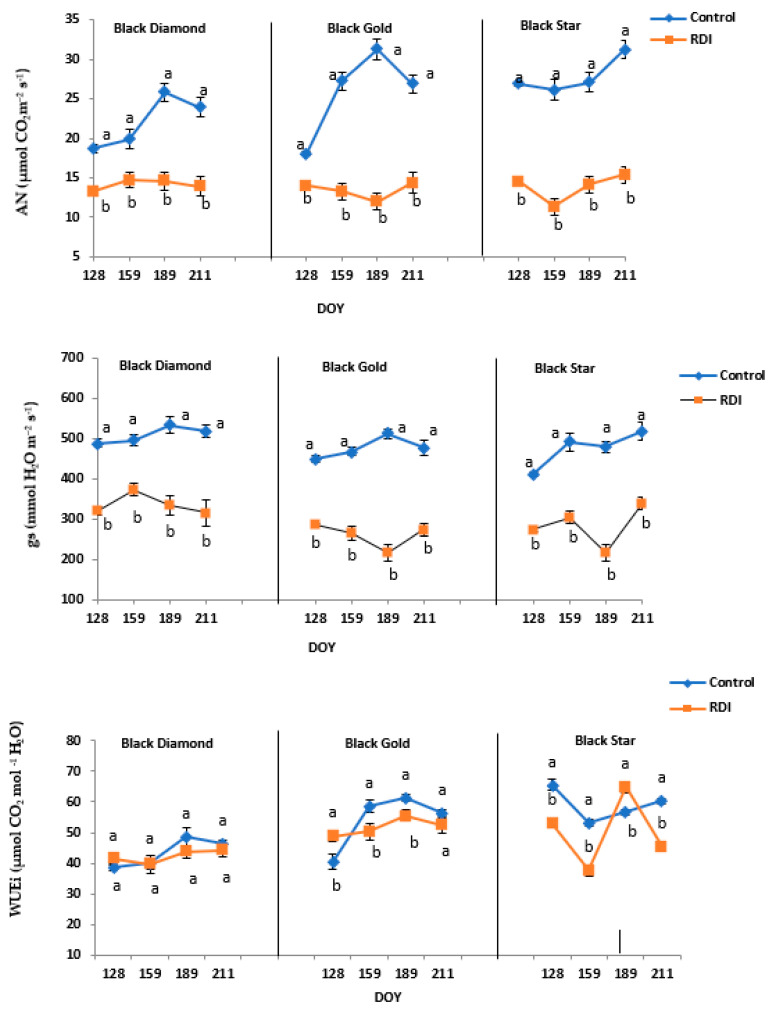
Modification of net assimilation photosynthetic rate (AN) (µmol CO_2_ m^−2^ s^−1^), stomatal conductance rate (gs) (mmol H_2_O m^−2^ s^−1^), and Intrinsic water use efficiency (WUEi = AN/gs) (µmol CO_2_ mol^−1^ H_2_O) of the three plum cultivars (BD, BG, and BS) subjected (RDI) or not (CI) to regulate deficit irrigation in a sunny day. Values are the means of twelve samples (*n* = 12) ± standard deviation. The letters a and b indicate significant differences (*p* ≤ 0.05), between the two irrigation treatments, within the same date.

**Figure 5 plants-11-01545-f005:**
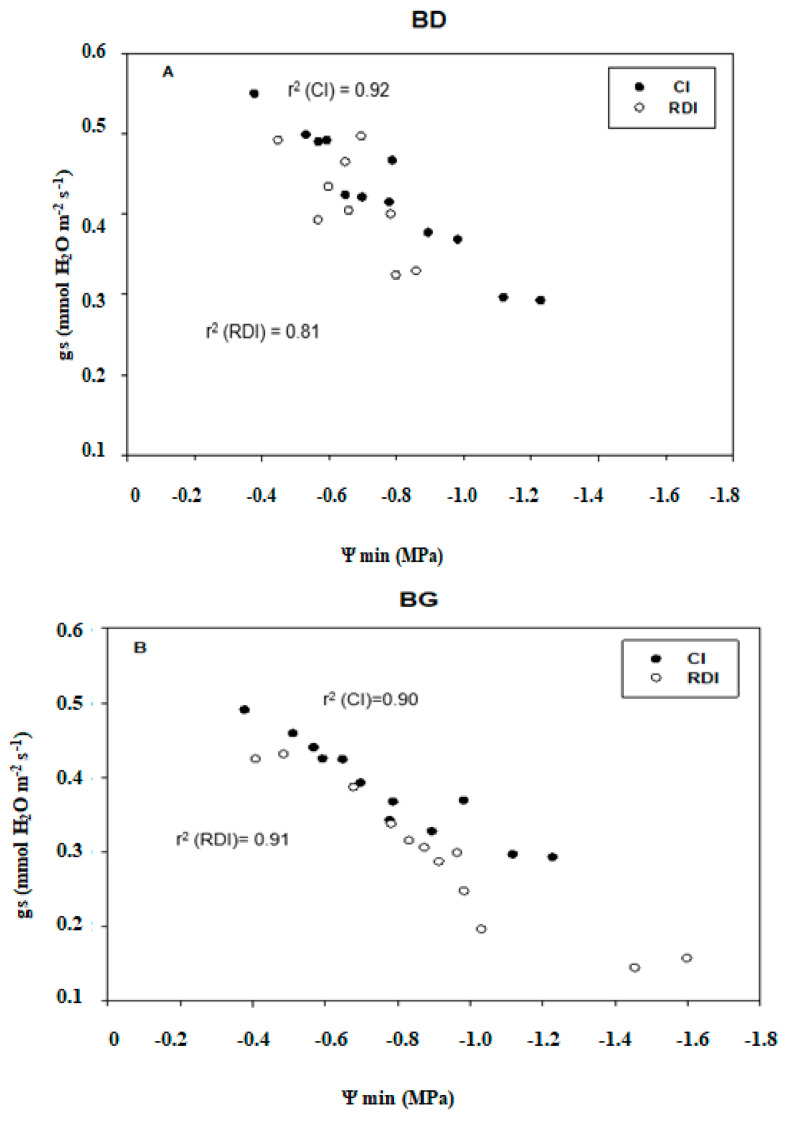
Relationship between stomatal conductance rate (gs) (mmol H_2_O m^−2^ s^−1^) and Predawn water potential (MPa) in three Black Diamond (**A**), Black Gold (**B**), and Black Star (**C**) cultivars in responses to both water treatments (CI and RCI). Values are the means of twelve samples (*n* = 12) ± standard deviation. r^2^ (CI), r^2^ for control irrigation; r^2^ (RDI), r^2^ for deficit irrigation.

**Figure 6 plants-11-01545-f006:**
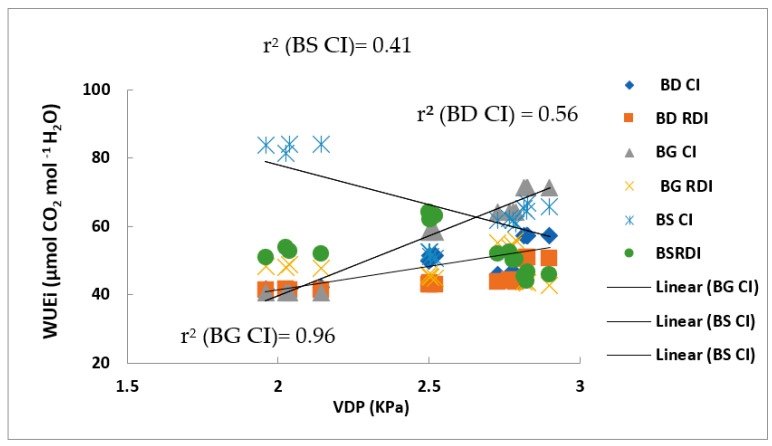
Intrinsic water use efficiency (WUEi = AN/gs) (µmol CO_2_ mol^−1^ H_2_O) and water vapour pressure deficit (KPa) in three Black Diamond, Black Gold, and Black Star cultivars in responses to both water treatments (CI and RDI). Each point is the average of three replications and vertical bars indicate ± SEM. r^2^ (CI), r^2^ for control irrigation; r^2^ (RDI), r^2^ for deficit irrigation.

**Table 1 plants-11-01545-t001:** Soil chemical composition in the studied orchards.

Depth (cm)	K_2_O (g kg^−1^)	CaO^+^ (g kg^−1^)	Na_2_O^+^ (g kg^−1^)	K_2_O/MgO	Fe^+^ (mg kg^−1^)	Mn^+^ (mg kg^−1^)	Cu^+^ (mg kg^−1^)	B (mg kg^−1^)	Zn^+^ (mg kg^−1^)
0–25	0.58	8.97	0.35	0.9	6	3.1	4.4	2.9	1
25–50	0.28	7.95	0.14	0.4	4	2	2.1	1.1	0.3

**Table 2 plants-11-01545-t002:** Effect of regulated-deficit irrigation (RDI) on Midday leaf water potentials (Ψmin (MPa)) and osmotic potentials (Ψos (MPa)) of three plum cultivars (Black Diamant, Black Gold, and Black Star) in the centre of Tun sia.

Cultivars	Treatments	Day of Years	Ψ min (MPa)	Ψ os (MPa)
**BS**	**Control**	128	−1.9 ± 0.08 ^a^	−0.63 ± 0.02 ^a^
		159	−2.2 ± 0.05 ^a^	−0.81 ± 0.04 ^a^
		189	−3.1 ± 0.05 ^a^	−0.91 ± 0.03 ^a^
		211	−2.2 ± 0.01 ^a^	−1.40 ± 0.04 ^a^
	**RDI**	128	−1.9 ± 0.03 ^a^	−0.83 ± 0.02 ^b^
		159	−2.6 ± 0.07 ^b^	−1.07 ± 0.01 ^b^
		189	−3.6 ± 0.02 ^b^	−1.16 ± 0.08 ^b^
		211	−3.1 ± 0.01 ^b^	−2.38 ± 0.07 ^b^
**BD**	**Control**	128	−2.3 ± 0.04 ^a^	−0.75 ± 0.09 ^a^
		159	−1.6 ± 0.03 ^a^	−0.87 ± 0.06 ^a^
		189	−2.1 ± 0.07 ^a^	−1.20 ± 0.01 ^a^
		211	−1.7 ± 0.09 ^a^	−1.51 ± 0.08 ^a^
	**RDI**	128	−2.3 ± 0.02 ^a^	−0.94 ± 0.07 ^b^
		159	−1.86 ± 0.07 ^b^	−1.20 ± 0.05 ^b^
		189	−2.5 ± 0.06 ^b^	−1.56 ± 0.02 ^b^
		211	−2.6 ± 0.05 ^b^	−2.22 ± 0.03 ^b^
**BG**	**Control**	128	−1.8 ± 0.02 ^a^	−1.61 ± 0.02 ^a^
		159	−1.8 ± 0.04 ^a^	−1.58 ± 0.05 ^a^
		189	−2.5 ± 0.05 ^a^	−1.56 ± 0.01 ^a^
		211	−2.1 ± 0.01 ^a^	−1.65 ± 0.01 ^a^
	**RDI**	128	−1.8 ± 0.02 ^a^	−2.21 ± 0.09 ^b^
		159	−2.3 ± 0.04 ^b^	−2.18 ± 0.08 ^b^
		189	−3.3 ± 0.07 ^b^	−2.32 ± 0.01 ^b^
		211	−2.6 ± 0.04 ^b^	−2.58 ± 0.05 ^b^

Each value is the mean of 3 replications. The letters a and b indicate significant differences (*p* ≤ 0.05) between the two irrigation treatments, within the same date. BD, Black Diamant; BG, Black Gold; BS, Black Star; CI, control irrigation; RDI, regulated deficit irrigation.

**Table 3 plants-11-01545-t003:** Effect of regulated-deficit irrigation (RDI) on fruit yield and pomological parameter of the fruits of three plum cultivars: BD: Black Diamant, BG: Black Gold, and BS: Black Star in the centre of Tunisia.

Varieties	Treatment	Fruit Weight (g)	Size (mm)	Yield (kg Tree^−1^)	Firmness (kg 0.5 cm^−2^)	Titratable Acidity (% Malic Acid)	SSC (°Brix)
BD	CI	96.33 ± 4.90 ^a^	59.24 ± 1.45 ^a^	62.5 ± 1.5 ^a^	5.22 ± 0.28 ^a^	3.96 ± 0.08 ^a^	16.66 ± 0.12 ^b^
	RDI	78.4 ± 2.14 ^b^	53.04 ± 0.69 ^b^	54 ± 2 ^b^	5.58 ± 0.68 ^a^	3.75 ± 0.03 ^b^	17.8 ± 0.15 ^a^
BG	CI	89.61 ± 2.82 ^a^	55.06 ± 0.67 ^a^	55 ± 1.8 ^a^	6.46 ± 0.11 ^b^	3.7 ± 0.09 ^a^	14.66 ± 0.06 ^a^
	RDI	77.87 ± 3.02 ^b^	52.30 ± 0.86 ^b^	45 ± 1.8 ^b^	7.57 ± 0.19 ^a^	3.54 ± 0.06 ^b^	14.93 ± 0.10 ^a^
BS	CI	104.57 ± 2.79 ^a^	57.41 ± 0.71 ^a^	67.5 ± 0.9 ^a^	2.11 ± 0.23 ^b^	3.65 ± 0.03 ^a^	15.13 ± 0.16 ^a^
	RDI	90.64 ± 3.99 ^b^	55.39 ± 0.56 ^a^	49.5 ± 1 ^b^	2.79 ± 0.03 ^a^	3.49 ± 0.01 ^b^	15.73 ± 0.16 ^a^

Each value is the mean of 3 replications. Different letters (a,b) indicate a significant difference between irrigation treatments. BD, Black Diamant; BG, Black Gold; BS, Black Star; CI, control irrigation; RDI, regulated deficit irrigation. The letters a and b indicate significant differences (*p* ≤ 0.05), between the two irrigation treatments, within the same date.

## Data Availability

Not applicable.

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
