# Peer review of "Effect of Regulated Deficit Irrigation on Agronomic Parameters of Three Plum Cultivars (Prunus salicina L.) under Semi-Arid Climate Conditions"

_plants, 2022, doi:10.3390/plants11121545_

Round 1
Reviewer 1 Report
Dear authors,
The paper entitled "Effect of regulated deficit irrigation on agronomic parameter of three plum cultivars (Prunus salicina L.) under semi-arid climate conditions" by Hichem Hajlaoui et al is an interesting study. I personally think that the topic approached could be of interest for the readers. The paper is generally well written and organized. I recommend that authors should check carefully the text for minor spell checks (starting with the title - Line 2- title- „paramter” – parameter; Line 102- „was to evaluated”- was to evaluate; Line 115- „the temperature was very high it varies between” – „the temperature was very high and varied between”). Also I would recommend that authors should review Figures 4 and 5 because something is not write in these 2 figures format/ the legend from left side of the page is unclear. The second figure from figure 5 should be redrawn because the formula/ value for r2 is incorrect written/introduced – a square appears instead of „=”).
I would recommend to introduce more correlation of your results with previous studies/literature in Discussion section (if possible and available from more recent years).
Author Response
Dear editor and reviewers,
Thanks for your valuable comments concerning our manuscript. Firstly, we would like to express our sincere appreciation for your professional remarks on our study. These comments are all valuable and very helpful for us to improve the quality of our study. We have studied comments item by item and we have made correction which we hope to meet with approval. The main revision in the study and detailed reply to reviewers are presented as follows:
You would find all modifications made in new version of our manuscript typed in red color.
Reviewer 1:
Dear authors,
The paper entitled "Effect of regulated deficit irrigation on agronomic parameter of three plum cultivars (Prunus salicina L.) under semi-arid climate conditions" by Hichem Hajlaoui et al is an interesting study. I personally think that the topic approached could be of interest for the readers. The paper is generally well written and organized. I recommend that authors should check carefully the text for minor spell checks.
Starting with the title – Line 2- title- „paramter” – parameter; Line 102
This was corrected in the revised version.
„was to evaluated”- was to evaluate;
This was corrected in the revised version.
Line 115- „the temperature was very high it varies between” – „the temperature was very high and varied between”).
This was corrected in the revised version.
Also I would recommend that authors should review Figures 4 and 5 because something is not write in these 2 figures format/ the legend from left side of the page is unclear. The second figure from figure 5 should be redrawn because the formula/ value for r2 is incorrect written/introduced – a square appears instead of „=”).
We correct the legend of these two figures in the revised version.
I would recommend to introduce more correlation of your results with previous studies/literature in Discussion section (if possible and available from more recent years).
We added more correlation with previous studies as suggested the reviewer. Please see line 407, 453, 458 and 476.

Reviewer 2 Report
Dear Authors,
I have found your work, entitled: "Effect of regulated deficit irrigation on agronomic paramter of three plum cultivars (Prunus salicina L.) under semi-arid climate conditions" interesting and important. You have done extensive work of importance in my opinion and I think your described information and conclusions could interest many researchers and readers. The thesis is a compendium of information with agronomical significance about the behavior of selected plum cultivars (Prunus salicina L.) in semi-arid climates under regulated deficit irrigation What is more, You created an extensive literature collection in this area, which is very beneficial for those interested in this topic. Although the work is interesting, I think that You should take a count modification of this article. I recommend publishing it in "Plants" after a small transformation.
General comments:
- Due to the fact that there are many abbreviations in the manuscript, it may be worthwhile to collect them in a dictionary of abbreviations.
- It is worth ordering the use of abbreviations according to the rule that when used for the first time in the text, it should be explained, later it does not have to be.The Authors are not faithful always to this principle and it causes a mess.
Introduction
Line 51- 56: please prove it by literature or some document
Line 74: …. Osmotic adjustment – osmotic adjustment?
Line 82 – 84: it is very interesting but should be more concrete and proved by more citations
Results
Line 132: Rainfall (pluv) and water vapour pressure deficit (VPD, KPa) in terms of DOY, average of daily mean air …. – should be: Rainfall (pluv) and water vapour pressure deficit (VPD (there is another abbreviation on the diagram, should be corrected), KPa) in terms of DOY (Day of Year), average of daily mean air ….
Line 133: … temperature (T) and crop reference evapotranspiration (ET0) in terms of DOY (Day of Year) measured for the period – should be: … temperature (T) and crop reference evapotranspiration (ET0) in terms of DOY (Day of Year) measured for the period
Line 172: Days of year – maybe standardize, as in Line 132?
Line 198: Table 2 …………………….. leaf water potentials (MPa) and osmotic potentials (MPa) – and in the table: „Ψ min Ψ os” – ???
Line 314 – 318: font for example CO2, H2O …
Discussion
Line 421 – 424: I do not understand why this data is not in the Results part as well as Figure 5 and Figure 6 with the part of text correlated with this Figure? In my opinion, it should be both transferred into the Results chapter.
Line 497: Prunus armeniaca L. – should be in Italic font: Prunus armeniaca L.
Results
Line 534: (figure.2) – should be: Figure 2:
Line 538: dots, please cancel one of them
Line 547 – 549: the names of the elements: potassium, calcium, magnesium should be in capital letters?
Line 563: The relative water content (RWC) was determined by the method of ………???
With best regards!
Author Response
Dear editor and reviewers,
Thanks for your valuable comments concerning our manuscript. Firstly, we would like to express our sincere appreciation for your professional remarks on our study. These comments are all valuable and very helpful for us to improve the quality of our study. We have studied comments item by item and we have made correction which we hope to meet with approval. The main revision in the study and detailed reply to reviewers are presented as follows:
You would find all modifications made in new version of our manuscript typed in red color.
Reviewer 2:
Dear Authors,
I have found your work, entitled: "Effect of regulated deficit irrigation on agronomic paramter of three plum cultivars (Prunus salicina L.) under semi-arid climate conditions" interesting and important. You have done extensive work of importance in my opinion and I think your described information and conclusions could interest many researchers and readers. The thesis is a compendium of information with agronomical significance about the behavior of selected plum cultivars (Prunus salicina L.) in semi-arid climates under regulated deficit irrigation What is more, You created an extensive literature collection in this area, which is very beneficial for those interested in this topic. Although the work is interesting, I think that You should take a count modification of this article. I recommend publishing it in "Plants" after a small transformation.
General comments:
Due to the fact that there are many abbreviations in the manuscript, it may be worthwhile to collect them in a dictionary of abbreviations. It is worth ordering the use of abbreviations according to the rule that when used for the first time in the text, it should be explained, later it does not have to be.The Authors are not faithful always to this principle and it causes a mess.
We added the abbreviation in the manuscript as asked the reviewer.
Introduction
Line 51- 56: please prove it by literature or some document
We added the reference to this section in the revised version.
Line 74: …. Osmotic adjustment – osmotic adjustment?
We mean a lowering of osmotic potential. This was corrected in the revised version.
Line 82 – 84: it is very interesting but should be more concrete and proved by more citations
We added more citation in the revised version.
Results
Line 132: Rainfall (pluv) and water vapour pressure deficit (VPD, KPa) in terms of DOY, average of daily mean air …. – should be: Rainfall (pluv) and water vapour pressure deficit (VPD (there is another abbreviation on the diagram, should be corrected), KPa) in terms of DOY (Day of Year), average of daily mean air ….
This was corrected in the revised version.
Line 133: … temperature (T) and crop reference evapotranspiration (ET0) in terms of DOY (Day of Year) measured for the period – should be: … temperature (T) and crop reference evapotranspiration (ET0) in terms of DOY (Day of Year) measured for the period.
This was corrected in the revised version.
Line 172: Days of year – maybe standardize, as in Line 132?
This was corrected in the revised version.
Line 198: Table 2 …………………….. leaf water potentials (MPa) and osmotic potentials (MPa) – and in the table: „Ψ min Ψ os” – ???
We added the abbreviation of lea water potential and osmotic water potential to the title of Table 2 in the revised version.
Line 314 – 318: font for example CO2, H2O …
This was corrected in the revised version.
Discussion
Line 421 – 424: I do not understand why this data is not in the Results part as well as Figure 5 and Figure 6 with the part of text correlated with this Figure? In my opinion, it should be both transferred into the Results chapter.
We agree with the comment of the reviewer. These parts were transferred to the result chapter in the revised version.
Line 497: Prunus armeniaca L. – should be in Italic font: Prunus armeniaca L.
This was corrected in the revised version.
Results
Line 534: (figure.2) – should be: Figure 2:
This was corrected in the revised version.
Line 538: dots, please cancel one of them
This was corrected in the revised version.
Line 547 – 549: the names of the elements: potassium, calcium, magnesium should be in capital letters?
This was corrected in the revised version.
Line 563: The relative water content (RWC) was determined by the method of ………???
We added the reference of the method of RWC in the revised version.
